# Influenza Vaccination Uptake and Hesitancy among Healthcare Workers in Early 2021 at the Start of the COVID-19 Vaccine Rollout in Cape Town, South Africa

**DOI:** 10.3390/vaccines10081176

**Published:** 2022-07-25

**Authors:** Samuel M. Alobwede, Elvis B. Kidzeru, Patrick D. M. C. Katoto, Evelyn N. Lumngwena, Sara Cooper, Rene Goliath, Amanda Jackson, Charles S. Wiysonge, Muki S. Shey

**Affiliations:** 1Department of Medicine, Faculty of Health Sciences, University of Cape Town, Cape Town 7925, South Africa; oliveasabo@gmail.com; 2Clinical Division, Department of Research and Innovation, Partners in Sexual Health, Cape Town 7500, South Africa; 3Centre for Research on Health and Priority Pathologies (CRHPP), Institute of Medical Research and Medicinal Plant Studies (IMPM), Ministry of Scientific Research and Innovation, Yaounde P.O. Box 13033, Cameroon; eb.kidzeru@uct.ac.za; 4Hair and Skin Research Laboratory, Division of Dermatology, Department of Medicine and Groote Schuur Hospital, University of Cape Town, Cape Town 7925, South Africa; 5Division of Radiation Oncology, Department of Radiation Medicine, University of Cape Town and Groote Schuur Hospital, Cape Town 7925, South Africa; 6Cochrane South Africa, South African Medical Research Council, Cape Town 7500, South Africa; katoto@sun.ac.za (P.D.M.C.K.); sara.cooper@mrc.ac.za (S.C.); charles.wiysonge@mrc.ac.za (C.S.W.); 7Centre for General Medicine and Global Health, Department of Medicine, University of Cape Town, Cape Town 7925, South Africa; 8School of Clinical Medicine, Faculty of Health Sciences, University of Witwatersrand, Johannesburg 2193, South Africa; en.lumngwena@uct.ac.za; 9Centre for the Study of Emerging and Re-Emerging Infections (CREMER), Institute of Medical Research and Medicinal Plant Studies (IMPM), Ministry of Scientific Research and Innovation, Yaounde P.O. Box 13033, Cameroon; 10School of Public Health and Family Medicine, University of Cape Town, Cape Town 7925, South Africa; 11Department of Global Health, Stellenbosch University, Francie Van Zijl Drive, Tygerberg, Cape Town 7505, South Africa; 12Wellcome Centre for Infectious Disease Research in Africa (CIDRI-Africa), Faculty of Health Sciences, University of Cape Town, Cape Town 7925, South Africa; rene.goliath@uct.ac.za (R.G.); amanda.jackson@uct.ac.za (A.J.); 13HIV and Other Infectious Diseases Research Unit, South African Medical Research Council, Durban 4091, South Africa; 14Institute of Infectious Disease and Molecular Medicine (IDM), Faculty of Health Sciences, University of Cape Town, Cape Town 7925, South Africa

**Keywords:** influenza vaccines, vaccine hesitancy, healthcare workers, South Africa

## Abstract

Vaccination attitudes among healthcare workers (HCWs) predict their level of vaccination uptake and intention to recommend vaccinations to their patients. To our knowledge, no study has been conducted in South Africa to assess hesitancy toward influenza vaccines among HCWs. We adapted a questionnaire developed and validated by Betsch and colleagues and used it to conduct online and face-to-face interviews among HCWs at the start of the COVID-19 vaccine rollout. Multivariate logistic regression was used to assess predictors of influenza vaccine hesitancy. Of 401 participants, 64.5% were women, 49.2% were nurses, and 12.5% were physicians. A total of 54.9% were willing to accept, 20.4% were undecided, and 24.7% intended to refuse influenza vaccination. Participants who were above 25 years of age and physicians were more likely to accept the vaccine. Key predictors of vaccine acceptance were confidence in the effectiveness, consideration of benefits and risks, and willingness to be vaccinated to protect others. Influenza vaccine hesitancy was highest in those who did not trust that influenza vaccines are safe. For future flu seasons, tailored education programs on the safety and effectiveness of flu vaccines targeting younger HCWs, could be vital to improving vaccine uptake.

## 1. Introduction

Influenza, known as the flu, is an acute respiratory infection that is highly contagious and considered one of the most challenging public health problems worldwide [1,2]. It may range from mild to severe illness, causing hospitalizations and deaths mainly among high-risk groups [1]. Seasonal influenza causes an estimated 3 to 5 million cases of severe illness and between 250,000 to 500,000 deaths worldwide every year [1]. Most often, deaths associated with influenza occur among the most vulnerable individuals, including young children, the elderly, and chronically ill patients [1]. Despite the severity of influenza, there are safe vaccines available, albeit with low uptake (i.e., vaccine hesitancy) among specific risk groups, which results in high burden of influenza infections. Healthcare workers (HCWs) are in regular contact with patients; therefore, they are at a higher risk of contracting the influenza virus, and may transmit the disease to their patients. As such, infections among HCWs may be a source of nosocomial outbreaks, with an increased risk of mortality among the immunocompromised, hospitalized, and high-risk patients they care for [3,4]. Furthermore, influenza infection among health professionals is associated with a high economic burden, mainly related to absenteeism [5].

The World Health Organization (WHO) identified vaccine hesitancy as one of the top ten threats to global health in 2019 [6,7,8]. The WHO, together with the U.S. Advisory Committee on Immunization Practices (ACIP), recommended mandatory annual vaccination of HCWs with influenza vaccines [9,10]. Despite these recommendations and the known efficacy of influenza vaccines in reducing infection and severity of sickness, HCW absenteeism from work due to flu remains high even in the high-income parts of the world [11,12]. This is associated with low annual vaccination rates; for example, in the UK, influenza vaccination coverage varies between 44% and 54% [11]. Vaccine coverage among healthcare professionals remains low mainly in low- and middle-income countries [13,14], with less than half having received the influenza vaccine at least once in their lives and only 15.3% vaccinated in the 2018–2019 flu season in Tunisia [14]. It is critical to understand barriers to influenza vaccination or factors associated with hesitancy and to find better ways to improve influenza vaccine uptake in this group, as they are on the front line [15].

The existing literature suggests that influenza vaccination influences COVID-19 infection, however, with inconsistent results across different studies. There are reports showing that influenza vaccination both protects against influenza and reduces the risk of COVID-19 infection, suggesting immune cross-protection between influenza vaccination and COVID-19 to an extent [16,17,18]. Therefore, increasing HCWs’ flu vaccination may provide benefits, especially during this period of relatively high vaccine hesitancy towards COVID-19 vaccination and especially among HCWs [19]. Moreover, there is no clarity on the association between influenza vaccination and COVID-19 infection. Kong and colleagues conducted a systematic review of 27 studies from 1 January 2019 to 31 December 2021 with the aim of evaluating the “effect of COVID-19 pandemic on influenza vaccination intention” [20]. Their findings from a systematic synthesis of 27 studies on the effects of the COVID-19 pandemic on influenza showed an increase in intention to vaccinate in 2020/2021, regardless of the participants’ demographics [20]. Thus, despite inconsistent results, this suggests that the COVID-19 pandemic increased the intention to vaccinate against influenza in other parts of the world.

We might wonder if this COVID-19 pandemic-driven increase in influenza vaccination influences SARS-CoV-2 infection. Another systematic review in 2020 by Del Riccio M and colleagues assessed the association between influenza vaccination and the risk of SARS-CoV-2 infection, severe illness, and death [21]. The result of another systematic synthesis found no evidence of a significant increase in risk of infection or in the severity illness or lethality, with several studies rather reporting significantly inverse associations [21]. Hence, these findings represent supporting measures aimed at raising HCWs influenza vaccination coverage in the COVID-19 era. In addition, in a recent study in 2022 by Kristensen and colleagues, they assessed in a prospective cohort study of 46,000 HCWs, asking whether their intention towards influenza vaccination had an impact on the risk of COVID-19 [22].

Further studies by Kristensen and colleagues found that influenza vaccination did not affect the risk of SARS-CoV-2 infection or COVID-19. In summary, the evidence thus far suggests that influenza vaccination does not increase risk of SARS-CoV-2 infection or of worse COVID-19 outcomes. Thus, understanding the impact of influenza vaccination on the risk of SARS-CoV-2 infection or COVID-19 on HCWs is critical, and would be of great importance to add to the evidence pool guiding public health policies in the future [23,24].

Hence, as an implementation research project on influenza vaccination among HCWs, we sought to find out whether HCWs would take the influenza vaccine during the next flu season. The study was conceived in March, when the flu season in South Africa is approaching (usually in winter between May and August) [25,26], which coincided directly with the rollout of COVID-19 vaccinations.

Vaccine hesitancy has been described by the Strategic Advisory Group of Experts on Immunization (SAGE) as a motivational state of being conflicted about or opposed to getting vaccinated, including intentions and willingness [27]. Vaccine hesitancy is a complex and context-specific issue, and may vary across time, place, and type of vaccine [28]. There are specific characteristics that should be considered when looking at influenza vaccine hesitancy. Influenza vaccine effectiveness varies yearly, and is sometime described as too low [28]; there are influenza-specific myths (e.g., the flu shot can cause the flu [29]); it is required yearly; and in most countries, it is recommended for specific risk groups only. Influenza vaccine hesitancy does have unique features that require further investigation in order to gain specific understanding of the phenomenon.

There are a limited number of studies on vaccine hesitancy, especially regarding influenza vaccines among HCWs conducted in low- and middle-income countries [30]. Prior to the COVID-19 pandemic there was a huge gap in knowledge about the attitudes of HCWs towards influenza vaccines as well as other vaccines in sub-Sahara Africa (SSA) and South Africa. Several studies have been reported in other parts of the world, including studies conducted in Canada, the UK, Saudi Arabia, Nigeria, and Ghana, finding a common theme whereby midwives, nurses, or physicians who reported accepting immunization for either Flu, HPV, or other conditions were those who trusted in their safety and efficacy and were thus more likely to recommend the vaccine to their peers and patients [31,32,33,34,35].

Concerns about the safety and adverse side effects of influenza vaccines were cited as main reasons for hesitancy in most studies [36]. This is supported by a study in which good knowledge of influenza vaccines, previous acceptance of vaccines, perception that the flu vaccine can decrease hospitalization, and prevention of influenza–COVID-19 co-infection had a positive influence on attitudes towards influenza vaccination [37]. In addition, there is evidence suggesting that HCWs’ lifestyle changes can shape their behaviour, and HCWs’ attitudes towards vaccination may be associated with changes in lifestyle. This was reported in a recent study in Italy where Gallé and colleagues assessed changes in public HCWs’ lifestyle during the COVID-19 pandemic era [38]. In a total of 1000 public HCWs that completed the study questionnaire, observed changes included that more than half of the HCWs had normal weight, were non-smokers, and slept at least 6 h per night. About a third consumed sweet foods and did not engage in physical activity.

However, little is known about the drivers of vaccination hesitancy among HCWs in Cape Town or the extent of its impact on vaccination coverage, as most research on vaccination hesitancy has been conducted in high-income countries [39,40]. Influenza vaccine hesitancy among HCWs remains a major public health challenge and has been understudied in sub-Saharan Africa compared to other parts of the world. To our knowledge, no study has been conducted in South Africa to assess acceptance and hesitancy of influenza vaccines among HCWs, especially during the rollout of COVID-19 vaccination in 2021, which was close to the winter period and flu season.

This study aimed to understand the extent and the major contributors to influenza vaccine acceptance and hesitancy among HCWs in Cape Town. These findings could be vital to understanding the bigger picture of the level of influenza vaccine acceptance and hesitancy among HCWs in South Africa as a whole.

## 2. Materials and Methods

### 2.1. Study Design and Setting

The study was a cross sectional survey of all HCWs 18 years and older of all races and genders living and working in hospitals and other healthcare settings in Cape Town, South Africa. We employed a convenience sampling and participants were HCWs from both private and government facilities who participated on a voluntary basis. These included nurses, physicians, pharmacists, hospital administrative personnel, health researchers, and radiologists. Our target population was HCWs in Cape Town who would have been enrolled or been eligible for the pilot phase of the COVID-19 vaccine rollout program in South Africa, which targeted frontline HCWs [41,42]. The sample size was estimated at 300 participants based on best practice recommendations for exploratory factor analyses and scale validation published by Betsch and colleagues in 2020 [43]. This sample size of 300 was enough to allow for detection of a reasonable correlation (r = 0.2) with about 95% power [43]. To assess the psychological antecedents of influenza vaccination in our study, we performed a qualitative assessment of confidence, complacency, constraints, calculated risk, and collective responsibility as determinants of influenza vaccine hesitancy amongst HCWs as previously described by Betsch and colleagues [43,44,45].

### 2.2. Ethical Consideration

The study was approved by the Human Research Ethics Committee of the University of Cape Town (HREC: 858/2020), and permission was granted to access health facilities in Cape Town by the Western Cape Provincial Government Department of Health (WC_202101_014). In addition, permission was granted by the respective healthcare facilities managers for data to be collected. Ethical principles were adhered to, including HCWs’ informed consent, anonymity, and confidentiality.

### 2.3. Data Collection and Management

Data collection started prior to the preceding flu season and coincided with the start of the COVID-19 vaccination rollout among HCWs in South Africa. Participants were enrolled between 15 March and 27 May 2021. Recruitment of participants and study data were collected concurrently either through hard copy questionnaires handed to the healthcare facilities, or using an online form on Research Electronic Data Capture software tools (REDCap; Vanderbilt University, Nashville, TN, USA) [46,47,48] hosted at the University of Cape Town server and sent through a link to HCWs’ e-mails or WhatsApp chat groups. After being completed electronically, data were directly captured and stored on REDCap. Data from the hard copy hand-filled questionnaires were transferred to the online form on the REDCap database by one researcher immediately after forms were collected from the healthcare facilities, then further verified by a second researcher. The survey instrument variables included sociodemographic characteristics as well as the attitudes and behaviors of HCWs towards influenza vaccines. Information on sociodemographic characteristics included date of birth, age, gender, education, healthcare worker’s role, religion, and personal income.

Regarding contributors to influenza vaccine hesitancy in HCWs, this study adapted fifteen standardized questions published by Betsch and colleagues in 2018 [49] on five psychological antecedents of vaccinations, namely, constraints, confidence, calculation of risk, complacency, and collective responsibility (known as the 5C tool) [49] and further contextualized in our setting. In addition, questions were asked regarding HCWs’ religion being compatible with influenza vaccination. The validation of the 5C tool and questionnaire for use in our setting is a subject of another manuscript currently in preparation. The detailed questionnaire employed in the study can be seen in Appendix A as well as in our previous publication [19]. Participants had to sign consent before attempting the questionnaires either as hard copies or the online form.

### 2.4. Description of Variables

The explanatory variables consisted of 24 questions or statements (six on sociodemographic characteristics and 18 on attitudes toward influenza vaccination).

We treated the sociodemographic variables as follows: age was transformed from a numerical to a categorical variable (18–24, 25–34, 35–44, 45–54, 55+) in order to examine differences between specific age groups. We grouped healthcare worker roles into administrative support and clinical researchers, nurses, physicians, and other HCWs to reflect the risk related to the function. We stratified the highest educational attainment into three categories: below high school, high school graduate, and university (from Bachelor to PhD). We grouped religion into Christian, Muslim, other religions (African Spirituality, Hindu, Buddhist, Jewish, etc.), and none. For personal income per month, categories included less than ZAR 10,000, between ZAR 10,000 and 50,000, and more than ZAR 50,000 at an exchange rate of 1 United States Dollar to 14 South African Rands.

Each of the 17 vaccine attitude statements had seven response options: 1 for strongly disagree, 2 for moderately disagree, 3 for slightly disagree, 4 for neutral, 5 for slightly agree, 6 for moderately agree, and 7 for strongly agree. We shortened these to three responses, as follows: 1 to 3 were categorized as “no”, 4 as “neutral”, and 5 to 7 as “yes”.

Our outcome variable was the intention to receive a dose of influenza vaccine. This was measured in the questionnaire by the statement “During the next influenza season, I will take the influenza vaccine”. This statement had seven response options: 1 for strongly disagree, 2 for moderately disagree, 3 for slightly disagree, 4 for neutral, 5 for slightly agree, 6 for moderately agree, and 7 for strongly agree. We transformed the seven responses to three as follows: 1 to 3 were categorised as “no” or “refusals”, 4 as “neutral” or “undecided”, and 5 to 7 as “yes” or “acceptance”. The outcome variable was further transformed to a binary variable, with responses 1 to 4 categorised as “hesitancy” and 5 to 7 as “acceptance”. This dichotomisation of Likert scale responses for influenza vaccine intention has been employed previously in other recent surveys [50,51], including our group’s recent assessment of attitudes and behaviours towards COVID-19 vaccination by HCWs [19].

### 2.5. Statistical Analyses

A total of 414 people completed the questionnaire. However, eight records were duplicates and five had incomplete information on age and gender. These records (13) were excluded from the statistical analyses. To describe the intentions of HCWs towards influenza vaccination by sociodemographic characteristics and by relevant personal considerations, frequencies and percentages were used for categorical data and means with standard deviation (SD) for continuous variables.

We used multivariate logistic regression to assess the association between sociodemographic variables and intention to receive an influenza vaccine dose, followed by models including each vaccination attitude variable separately. We assessed the following predictors: influenza vaccine acceptance versus hesitancy (i.e., neutral and refusal); acceptance versus neutral, excluding the refusals; acceptance versus refusal, excluding neutrals; neutral versus refusal, excluding acceptance; and refusal versus other intentions (i.e., acceptance and neutral). As a result of this process, we fitted models adjusted for age as a continuous variable and gender in order to assess independent predictors of influenza vaccine intention among HCWs in Cape Town. The strength of the association was expressed by crude and adjusted odds ratios (OR) and accompanying 95% confidence intervals (CI). The reported *p*-values are exact and two-tailed, with a value of 0.05 deemed statistically significant.

All statistical analyses and tables were processed with R software version 4.0.4, and all models were built using the generalised linear model for logistic regression in the R/finalfit R 1.0.3 package.

## 3. Results

### 3.1. Sociodemographic Characteristics of Study Participants

Table 1 presents the sociodemographic characteristics of 401 participants of this study as we investigated the intention to take the vaccine. Of the 401 HCWs included in the analysis, 220 (54.9%) expressed an intention to take the influenza vaccine, 99 (24.7%) an intention to refuse, and 82 (24.4%) were neutral. Looking at age, participants between the ages 25–34 and 35–44 were more willing to take the influenza vaccine; both age groups had a participation frequency of 104 (25.9%) and 103 (26.7%), with an intention to accept influenza vaccination frequency of 56 (25.5%) and 58 (26.4%), respectively. Professional classification found the highest number of HCWs that participated in this study to be nurses (49.2%), with physicians the lowest number (12.5%). Among the HCWs who expressed their intention to take the shot, 135 (64.3%) female HCWs declared their intention to accept the influenza vaccine. Among all HCWs, 193 nurses (49.2%) participated in this study, with 86 (40.2%) expressing their intention to be vaccinated for influenza, while 56 (68.3%) were neutral and 51 (53.1%) refused. More high school and university graduates intended to be vaccinated, with acceptance frequencies of 86 (40.6%) and 118 (55.7%), respectively, among all HCWs. In addition, those with a personal income of between ZAR 10,000–50,000 per month had the highest participation frequency of 235 (64.2%), 59.1% of whom expressed their intention to accept the flu vaccine. Regarding religion, 262 (71.6%) Christians participated, and 73.3% expressed an intention to accept the influenza vaccines.

### 3.2. Attitudes towards Influenza Vaccination among HCWs

We found that 49.9% (200 participants) of HCWs had received an influenza vaccine in the past (Table 2). From the 401 total participants who were included in the statistical analysis, 220 (54.9%) were willing to take the vaccine, with influenza vaccine hesitancy at 45.1%, i.e., a combination of those who were undecided and those who would refuse to take the flu vaccine. A total of 167 (76.6%) of the 220 who would accept the flu vaccine agreed that vaccination against influenza is compatible with their religion.

Based on confidence [43,44,45], out of the 220 participants that were willing to accept flu vaccination, 188 (86.2%) HCWs believed that the influenza vaccine is safe as compared to 11 (5.05%) that did not believe this. Moreover, 188 (87.0%) agreed that the influenza vaccine is effective. Of interest was that 178 (82.0%) participants expressed confidence that public authorities had the best interest of the general community at heart.

As for complacency [43,44,45], among the 401 participants included in the statistical analysis, 252 (64.8%) had the belief that influenza vaccination is not necessary because flu is not common anymore. Among the 220 who intended to take the flu vaccine, 162 (75.7%) believed the flu vaccine was necessary and 21 (9.81%) believed it was not. In addition, among the 401 participants included in the statistical analysis, 210 (54.0%) did not agree that their immune system is strong enough to protect them against influenza. Among the 220 who were willing to accept the influenza vaccine, 134 (62.6%) did not belief in the strength of their immune system to protect them, while 46 (21.5%) did. Furthermore, among the 220 HCWs who were willing to accept the influenza vaccine, 147 (68.1%) had the belief that the flu infection is severe enough that they should take a vaccine.

Regarding constraints [43], out of the 401 participants included in the statistical analysis, 264 (67.0%) of HCWs did not have the impression that everyday stress would prevent them from getting vaccinated against influenza. Among the 220 who intended to take the flu vaccine, 166 (76.1%) did not have the impression that everyday stress would prevent them from getting vaccinated against influenza, while 25 (11.5%) did. On the other hand, a smaller frequency of HCWs 76 (19.2%) out of the 401 participants included in the final analyses thought it would be inconvenient for them to get vaccinated against influenza. In addition, with respect to convenience [43,44,45], few HCWs, 89 (22.7%) out of the 401 participants included in the statistical analysis, thought that visiting the vaccination clinic would make them feel uncomfortable and that this would further keep them from getting vaccinated against influenza. However, 237 (60.5%) did not think this, of which 156 (72.9%) out of the 220 who intended to take the flu vaccine did not.

Looking at risk calculation [43,44], among the 220 who intended to take the flu vaccine, 149 (68.0%) indicated they would first weigh benefits and risks before taking the decision to receive the influenza vaccine. In addition, 144 (67.0%) HCWs highlighted that they would closely monitor whether or not the vaccine was useful to them when making the decision to accept vaccination. Importantly, a high frequency of HCWs 164 (75.9%) reported that it was of importance for them to fully understand the topic of vaccination before receiving the influenza vaccine.

As far as collective responsibility is concerned [43], among the 220 who intended to take the flu vaccine, 144 (67.6%) HCWs mentioned that even if the greater population is vaccinated for influenza, they need to get vaccinated as well. A majority of HCWs, 174 (80.9%), indicated that they would get vaccinated to protect their colleagues, patients, and other people with weaker immune systems. Finally, 184 (86.4%) HCWs reported that they believed vaccination to be a collective responsibility to prevent the spread of diseases such as influenza.

### 3.3. Predictors of Acceptance versus Hesitancy of Influenza Vaccination

HCWs’ willingness to accept influenza vaccination was significantly associated with age, with older HCWs (>24 years age groups) being more willing compared to the 17–24 years old age group. The older HCWs, 55–78 years old, were the most willing to accept the influenza vaccine (OR 9.53, 95% confidence interval (CI) 2.55–40.32, *p* = 0.001, Table 3). This was followed by HCWs between the ages of 45–54 years (OR 6.54, 95% CI 2.26 to 21.27, *p* = 0.001) and 35–44 years (OR 4.27, 95% CI 1.60 to 12.93, *p* = 0.006), and lastly 25–34 years old’s (OR 3.31, 95% CI 1.25 to 9.97, *p* = 0.02). Compared to HCWs with administration roles, nurses had the highest number of participants in the study, with a trend of significant unwillingness to accept the influenza vaccine (OR 0.48, 95% CI 0.21–1.09, *p* = 0.083), while physicians displayed a significantly higher willingness to accept the influenza vaccine (OR 24.43, 95% CI 3.97–480.39, *p* = 0.004). In addition, HCWs who attained higher education or university graduation had a significantly higher willingness towards influenza vaccination (OR 1.83, 95% CI 1.01–3.37, *p* = 0.048) than those with high school level. Among our study participants, gender, personal income, and religion did not prove to be independent predictors of influenza vaccine hesitancy.

Furthermore, when considering confidence in influenza vaccines, we noted that participants who had received an influenza vaccine in the past were more willing to take the vaccine (OR 6.41, 95% CI 3.15–13.53, *p* < 0.001) compared to those who had not previously received the vaccine. In addition, HCWs who believe that influenza vaccines are effective were more likely to accept the vaccines, with an acceptance frequency of 85.1% versus 14.9% hesitancy (OR 5.82, 95% CI 1.85–18.86, *p* = 0.003) compared to those with a neutral position.

Regarding variables related to complacency, 64.3% of HCWs were likely to take the influenza vaccine and agreed that vaccination was necessary (OR 3.58, 95% CI 1.83–7.22, *p* < 0.001). Furthermore, 63.8% of HCWs with a significant likelihood to accept influenza immunization did not think that their immune system was strong enough to protect them from flu (OR 2.05, 95% CI 1.06–3.99, *p* = 0.034).

When looking at aspects of constraints as predictors of influenza vaccination among HCWs, 62.9% of HCWs were more likely to accept vaccines for influenza and did not believe that everyday stress was a hindrance (OR 2.05, 95% CI 1.02–4.18, *p* = 0.046), while 65.8% of participants were willing to be vaccinated against influenza and did not agree that clinic visits were uncomfortable for them (OR 3.55, 95% CI 1.78–7.27, *p* < 0.001).

It is worth noting that 60.5% (OR 2.80, 95% CI 1.42–5.65, *p* = 0.003) of HCWs would consider the usefulness (benefits) of each dose of the influenza vaccine before deciding, and 63.6%, (OR 2.97, 95% CI 1.20–7.58, *p* = 0.020) did not think it was important to fully understand the topic of vaccination before getting vaccinated.Finally, collective responsibility is of great importance as a predictor of vaccine acceptance. There were a significantly higher number of HCWs (78.7%) with a higher likelihood of accepting influenza vaccination who believed that even when most people are vaccinated against influenza, they themselves must still be vaccinated (OR 6.96, 95% CI 3.24–15.50, *p* < 0.001). On the same note, influenza vaccine acceptance was significantly higher (68.0%) among participants who wanted to be vaccinated to protect people with weaker immune systems (OR 3.44, 95% CI 1.56–7.95, *p* = 0.003). In addition, 63.4% of HCWs who considered vaccination as a collective responsibility to prevent the spread of influenza had a significantly higher likelihood of receiving the influenza vaccine (OR 2.87, 95% CI 1.23–7.07, *p* = 0.02).

## 4. Discussion

There is huge gap in sub-Sahara Africa, and South Africa in particular, when it comes to research on influenza vaccine hesitancy among HCWs. Vaccine hesitancy represents a motivational state of being conflicted or opposed to vaccination [52]. This study was strategically designed to bridge this gap and to understand the behavior of HCWs towards influenza vaccine uptake, as this may influence the community’s response. It was initially reported that influenza vaccination had a protective effect or reduced the severity of COVID-19 infection. In this study, vaccine acceptance (willingness to vaccinate) was 54.9% while vaccine hesitancy (intention to refuse or undecided) was present among 45.1% of HCWs. The study findings reveal that the key drivers of influenza vaccine uptake among HCWs in Cape Town were confidence in the effectiveness of influenza vaccines and the collective responsibility and necessity of receiving the influenza vaccine in the next flu season. Another important factor was that, in addition to vaccination as a collective responsibility, 65.5% of HCWs had the belief that vaccination was essential to protecting their co-workers, members of the community, or patients with weaker immunity.

In addition, constraints as one of the predictors turned out to be a key factor considering vaccine hesitancy in HCWs. We showed that HCWs who did not believe that everyday stress had an impact on the decision to vaccinate in the next flu season had a strong likelihood of accepting influenza vaccination. Other vital findings of the study include that positive perceptions as to the efficacy, safety, trust, and need to understand the science behind vaccine development were strong predictors of acceptance. Other determinants of vaccine acceptance were demographic characteristics (e.g., age, religion, gender, education, and personal income), which showed varying degrees of association with vaccination acceptance and uptake. Government trust was associated with vaccine acceptance, and this was in line with the government’s intention to attain herd immunity.

This study is of critical importance considering that HCWs are the custodians of global health care. The WHO in 2012 and 2016 updated its recommendation on influenza vaccination of HCWs, indicating that ‘‘HCWs are an important priority group for influenza vaccination, not only to protect the individual and maintain healthcare delivery during influenza epidemics, but also to limit spread of influenza to vulnerable patient groups” [53,54]. The WHO recommended that annual influenza vaccination among HCWs be mandatory in order to mitigate the epidemiological and economic effects of seasonal influenzas [55,56,57]. However, influenza vaccine hesitancy among HCWs remains a major public health challenge, and has been understudied or neglected in sub-Saharan Africa compared to other parts of the world.

This study found that the overall coverage rate of influenza vaccination among HCWs in Cape Town, South Africa based on past behavior towards influenza vaccination was 49.9%, and the difference between those who were most likely to accept vaccination in the coming flu season (77.5%) and those who would not (22.5%) was significant. It is worth noting that a vaccine coverage of 49.9% in a mixed setting of health facilities that includes physicians, nurses, biomedical scientists, allied HCWs, hospital administrators, and others is quite low, and comparable to that of the only other extant African study on hesitancy towards influenza vaccination among HCWs in Tunisia, which was below 50% [14]. Cherif and colleagues reported that less than half of the health professionals enrolled, i.e., 36.6% had received an influenza vaccine at least once in the past, and only 15.3% were vaccinated against influenza in the 2018–2019 influenza season when the survey was conducted [14]. In line with our findings, less than half of HCWs (49.9%, which is higher than the findings of Cherif et al.) reported having received a shot of influenza vaccine in the past, while on the contrary more than half (54.9%) in our study were willing to accept influenza vaccination in the coming flu season.

Interestingly, in the Tunisian study participants with a high educational level were less likely to receive the influenza vaccine than those with the lowest educational level [14]. This was the contrary in South Africa, as we recorded a high rate of HCWs willing to accept influenza vaccination who were university (55.7%) or high school (40.6%) graduates, similar to previous results from a study in Italy [12]. This further emphasises the importance of tailored education programs targeting HCWs generally in Africa, following from previously low vaccination rates below expectations among both Tunisian and South African HCWs. A key observation between our studies is that most HCW participants in Tunisia were female (80%), while we recorded 64.5% female participants; while about 15% less, both studies had a majority of female participants.

This study was conducted at the peak of the COVID-19 pandemic, and recruitment of HCWs in the study coincided with the campaigns of the COVID-19 vaccine rollout to HCWs in South Africa. This provides a better understanding of factors affecting influenza vaccination among HCWs in the next and subsequent flu seasons during the COVID-19 pandemic. Another study during the COVID-19 pandemic in Saudi Arabia assessing vaccination trends from 2017 to 2020 with 424 HCWs as participants had the majority being nurses (72.2%), with physicians making up 27.8% [32]. Of note is the similar sample size to our study, as we included 401 HCWs in our analysis, and the majority of participants in our study were nurses (49.2%) against 12.5% physicians, which was the lowest of all HCW roles in our study.

Previous reports have indicated an increase in influenza vaccine uptake from 2017 to 2019 (45%, 52% and 62%) and a decrease in 2020 (59%) in the flu season during the COVID-19 pandemic. This could be because of conspiracy theories on social media and other communication platforms, lack of proper communication between governments and other community stakeholders, and most importantly, complacency and inadequate education of HCWs as to the importance of their being vaccinated to protect their patients and community. Jones and colleagues in 2020 suggested that measures to control the COVID-19 pandemic might be quashing the cold and flu season [58], which could influence HCWs’ choice towards influenza vaccination. Following multivariate logistic regression, HCWs above 40 years of age, female, nurses, and participants who were knowledgeable about flu vaccination had a higher likelihood of having received the influenza vaccine and were more willing to accept vaccination in the next flu season in 2021 [32].

Our findings of an overall influenza vaccination rate of 49.9% were similar those of Alkathlan and colleagues during the COVID-19 pandemic [32]. In addition, following multivariate logistic regression, our analysis shows a very strong likelihood of older HCWs 35–44 years and above accepting the influenza vaccine in the next flu season, with the likelihood growing stronger in an age group-dependent manner. In addition, considering the usefulness of each vaccine prior to acceptance of flu vaccination showed a strong association with the likelihood of HCWs accepting vaccination in the next flu season. However, gender was not a strong enough variable to influence HCWs’ decision to vaccinate or not. On the contrary, we showed that physicians were most likely to receive influenza vaccination in the next flu season, while nurses showed a trend towards a strong likelihood of vaccination. In addition, HCWs who did not believe that knowledge about vaccination was necessary for influenza vaccine uptake had a strong likelihood of accepting vaccination in the next flu season.

Previous findings from Saudi Arabia [32] were in accordance with another study carried out in Lebanon during the COVID-19 pandemic in 2020 on 560 Lebanese HCWs, who had the belief that good knowledge of vaccination, having previously received the influenza vaccine, perception of flu vaccine benefits in decreasing hospitalization, and prevention of influenza–COVID-19 co-infection had a positive influence on attitudes towards influenza vaccination [59]. Kong et al. conducted a meta-analysis and systematic review of 27 studies from 1 January 2019 to 31 December 2021 with the aim of evaluating the “effect of COVID-19 pandemic on influenza vaccination intention” [20]. Their findings showed that studies reported increased intention to vaccinate in 2020/21, regardless of demographics. In accordance with our study, HCWs who believed that influenza vaccines are effective were more likely to accept the vaccines compared to those with neutral positions and there is evidence of increasing influenza vaccination coverage rates during the COVID-19 pandemic [24]. In addition, older HCWs were most likely to receive influenza vaccines compared to younger colleagues. Overall, this could be supported by the fact that HCWs’ confidence in influenza vaccines would lead to them recommending the vaccine to their co-workers, patients, or patients’ families compared to those who did not intend to be vaccinated, as reported in several studies carried out in Canada [31], the UK [33], and Israel [34].

Notably, in our study, the strongest drivers of vaccine uptake by HCWs were confidence in the effectiveness of influenza vaccines (85.1%) and belief that influenza vaccination was necessary (64.3%). This was supported by strong likelihoods after logistic regression of willingness to receive the influenza vaccine. This is in accordance with previous studies in China and Lebanon which reported that attitudes toward influenza vaccination were the strongest predictor of HCWs’ intention, actual acceptance, and recommendation status with respect to influenza vaccination [37,60]. The findings from our study can be explained in light of findings in a study by Costantino and colleagues, performed in three Sicilian University Hospitals, assessing influenza vaccination adherence or refusal and the attitudes and perceptions of vaccinated HCWs during the 2019/2020 influenza season [61]. They reported that out of a total of 2356 vaccinated HCWs that answered their questionnaire, the main reason for influenza vaccination adherence was to protect patients. Furthermore, higher self-perceived risk of contracting influenza and a positive attitude towards recommending vaccination to patients were significantly associated with influenza vaccination adherence during the last five seasons as reported after multivariable analysis. They found that fear of an adverse reaction was the main reason for influenza vaccine refusal. One key difference between both studies was the larger sample size in the study in Italy, compared to 414 in our study. In addition, several studies conducted in Canada [31], Ghana [36], the UK [33], Israel [34], and Lebanon [37] have reported that HCWs’ confidence in accepting influenza vaccines led to them recommending the vaccine to their co-workers, patients, or patients’ families compared to those who did not intend to be vaccinated.

On the contrary, HCWs in studies carried out in Greece and Costa Rica reported that concern about the effectiveness of flu vaccines was the most common barrier among other factors to flu vaccination [62,63]. However, these studies reported low, suboptimal, and decreasing influenza vaccine coverage in HCWs, despite their positive attitudes toward the influenza vaccine.

In addition, constraints turned out to be a key factor considering vaccine hesitancy in HCWs. We showed that HCWs who did not believe that everyday stress had an impact on decision-making regarding vaccination in the next flu season had a strong likelihood of accepting influenza vaccination. This was in line with a study carried out in Malaysia, where more than half (56.2%) of HCWs reported that time constraints were the most common reason for not having the vaccine [64]. Protecting themself was the most common reason reported for vaccination against influenza infection (73.6%), and 85.3% of respondents had the belief that influenza vaccination was important for disease prevention.

The findings of this study were in support of HCWs’ confidence in the effectiveness of influenza vaccines in preventing disease, with a strong association with likelihood of accepting vaccination in the next flu season. In addition, HCWs perceived vaccination as a collective responsibility and action, resulting in a strong likelihood of accepting vaccination in the next flu season. Another important factor was that, in addition to vaccination as a collective responsibility, 65.5% of HCWs had the belief that vaccination was essential to protect their co-workers, members of the community, and patients with weaker immunity. Following logistic regression, we showed that there was a stronger likelihood of HCWs’ intention to receive the flu vaccine in the next flu season. This was in accordance with studies carried out in Oman on 390 HCWs and in Greece on 363 HCWs, who responded that their main reason for vaccine acceptance was to protect themself, their family, patients, colleagues, and the community at large [62,65].

This study had several limitations. We might not have exhausted all the important questions concerning HCWs’ attitudes, knowledge, and practices regarding influenza vaccination. It is possible that the lower influenza rate during the pandemic was due to (1) the fact that influenza rates were lower during the pandemic, and were therefore not considered a priority; (2) the fear of visiting medical clinics due to the possibility of contracting COVID-19; or (3) the fact that a new respiratory virus infection was instead becoming a priority. Future research may explore correlating the number of monthly influenza cases with vaccination hesitancy, as well as analysing vaccine hesitancy in a prospective cohort accounting for high vs. low community COVID-19 transmission periods.

HCWs’ attitudes towards vaccination may be associated with changes in lifestyle (such as weight, diet, physical activity, and sleep), as shown in a recent study in Italy by Gallé and colleagues assessing changes in public HCWs’ lifestyle during the COVID-19 pandemic [38]. However, we did not test for the impact of HCWs’ lifestyle changes with respect to influenza vaccines at the start of the COVID-19 vaccine rollout in Cape Town. Moreover, the shortcomings of this study include the language of the questionnaire being limited to English, without the ability to consider the two other official frequently-spoken languages in Cape Town, Afrikaans and IsiXhosa. Finally, while informative, because of the use of convenience sampling and the disparities that may occur between provinces these findings may not be generalized to the national population. We believe that, although we had a justifiable sample size to provide enough statistical power, a larger sampling would probably have strengthened our observation. The third limitation is that this was a cross-sectional survey with data collected at a single time point, and there is a possibility that HCWs’ attitudes about influenza vaccination might change over time.

## 5. Conclusions

This study has shown that interventions that enhance knowledge and accessibility about influenza vaccines are warranted in order to improve vaccination coverage among HCWs. Influenza vaccine coverage was below 50%, demonstrating vaccine coverage that is suboptimal and below expectations. Hence, governments may look for ways and strategies to encourage influenza vaccination among HCWs. Key drivers of vaccine uptake in HCWs include confidence in the effectiveness of influenza vaccines and the collective responsibility and necessity of receiving the influenza vaccine even if everyone is vaccinated in order to protect the vulnerable, elderly, and those with poor immune systems. This reaffirms the importance of building on the confidence of HCWs about flu vaccines and their desire to accept vaccination in future. Key drivers of vaccine hesitancy included age, particularly among younger HCWs, and the desire to consider each vaccine dose before acceptance. The study was conducted using a cross-sectional survey approach and with an analysed sample size of 401. We recommend that more research on influenza vaccine hesitancy in HCWs be carried out in other parts of South Africa in order to determine whether vaccination uptake and hesitancy is a general trend. In addition, we recommend that future studies include questions that address HCWs’ lifestyle (such as weight, diet, physical activity, and sleep), as these may be associated with attitudes towards vaccination, as shown in a recent study in Italy by Gallé and colleagues assessing changes in public HCWs’ lifestyle during the COVID-19 pandemic era [38].

For future flu seasons, the importance of tailored education programs targeting younger HCWs, continuing heath education, and campaigns about vaccines uptake providing more information about the content of flu vaccines are vital to improving vaccine uptake. In addition, emphasis on inclusion of education about vaccines and viral diseases in the medical curriculum as one approach to mitigating vaccine hesitancy among HCWs should be encouraged. This would help to alleviate fears and provide avenues to engage and assist communities in addressing the pertinent issues and building influenza vaccine confidence among HCWs to reassure them about the effectiveness and safety of the seasonal flu vaccines. Finally, we believe that this paper provides scientific evidence on influenza vaccine acceptance among HCWs in South Africa and in the broader sub-Saharan African region. Furthermore, it provides a better understanding of influenza vaccine hesitancy among HCWs during the time of the COVID-19 pandemic.

## Figures and Tables

**Table 1 vaccines-10-01176-t001:** Sociodemographic features of 401 healthcare professionals in Cape Town, South Africa, stratified by their intention to accept, decline, or remain neutral once the influenza vaccination is available to them during COVID-19 pandemic. Survey period coincides with the start of COVID-19 vaccine rollout.

Variables	Intention for Influenza Vaccination			
	All	Neutral	Refusal	Acceptance
	N = 401	N = 82 (20.4%)	N = 99 (24.7%)	N = 220 (54.9%)
Sociodemographic Variables				
Age in years	39.2 (12.3)	33.6 (9.34)	38.2 (12.9)	41.7 (12.3)
Age groups:				
18–24 years	40 (9.98%)	15 (18.3%)	16 (16.2%)	9 (4.09%)
25–34 years	104 (25.9%)	29 (35.4%)	19 (19.2%)	56 (25.5%)
35–44 years	103 (26.7%)	22 (26.8%)	22 (22.2%)	58 (26.4%)
45–54 years	67 (16.7%)	8 (9.76%)	18 (18.2%)	41 (18.6%)
55–78 years	42 (10.5%)	1 (1.22%)	8 (8.08%)	33 (15.0%)
Gender:				
Male	136 (35.5%)	20 (25.6%)	41 (43.2%)	75 (35.7%)
Female	247 (64.5%)	58 (74.4%)	54 (56.8%)	135 (64.3%)
Healthcare worker role:				
Admin support	51 (13.0%)	8 (9.76%)	11 (11.5%)	32 (15.0%)
Nurses	193 (49.2%)	56 (68.3%)	51 (53.1%)	86 (40.2%)
Other health workers	99 (25.3%)	17 (20.7%)	32 (33.3%)	50 (23.4%)
Physicians	49 (12.5%)	1 (1.22%)	2 (2.08%)	46 (21.5%)
Highest educational level attained:				
High School Graduate	188 (49.3%)	47 (61.0%)	55 (59.8%)	86 (40.6%)
Below High School	11 (2.89%)	0 (0.00%)	3 (3.26%)	8 (3.77%)
University	182 (47.8%)	30 (39.0%)	34 (37.0%)	118 (55.7%)
Personal income:				
Less than R10,000 per month	78 (21.3%)	15 (18.8%)	25 (28.4%)	38 (19.2%)
More than R50,000 per month	53 (14.5%)	5 (6.25%)	5 (5.68%)	43 (21.7%)
R10,000–R50,000 per month	235 (64.2%)	60 (75.0%)	58 (65.9%)	117 (59.1%)
Religion:				
African Spirituality	4 (1.09%)	1 (1.25%)	0 (0.00%)	3 (1.54%)
Buddhist or Hindu	15 (4.10%)	5 (6.25%)	4 (4.40%)	6 (3.08%)
Christian	262 (71.6%)	55 (68.8%)	64 (70.3%)	143 (73.3%)
Jewish	4 (1.09%)	0 (0.00%)	1 (1.10%)	3 (1.54%)
Muslim	54 (14.8%)	17 (21.2%)	16 (17.6%)	21 (10.8%)
None	27 (7.38%)	2 (2.50%)	6 (6.59%)	19 (9.74%)

Values shown are absolute counts (percentages), except for age in years where the values are means (standard deviations).

**Table 2 vaccines-10-01176-t002:** Attitudes toward influenza vaccination of 401 healthcare professionals in Cape Town, South Africa, using the 5C questionnaire items (with the inclusion of religious influence), stratified by their intention to accept, decline, or remain neutral once influenza vaccine is available to them during COVID-19 pandemic. Survey period coincides with the start of COVID-19 vaccine rollout.

Variables	Intention to Get Influenza Vaccine			
	All	Neutral	Refusal	Acceptance
	N = 401	N = 82 (20.4%)	N = 99 (24.7%)	N = 220 (54.9%)
Received influenza vaccine in the past	200 (49.9%)	15 (18.3%)	30 (30.9%)	155 (72.8%)
Influenza vaccination is compatible with my religion:				
Neutral	86 (21.6%)	51 (62.2%)	11 (11.2%)	24 (11.0%)
No	100 (25.1%)	15 (18.3%)	58 (59.2%)	27 (12.4%)
Yes	212 (53.3%)	16 (19.5%)	29 (29.6%)	167 (76.6%)
I am completely confident that influenza vaccines are safe:				
Neutral	92 (23.2%)	57 (71.2%)	16 (16.2%)	19 (8.72%)
No	83 (20.9%)	11 (13.8%)	61 (61.6%)	11 (5.05%)
Yes	222 (55.9%)	12 (15.0%)	22 (22.2%)	188 (86.2%)
Influenza vaccination is effective:				
Neutral	94 (24.1%)	57 (71.2%)	21 (22.3%)	16 (7.41%)
No	75 (19.2%)	9 (11.2%)	54 (57.4%)	12 (5.56%)
Yes	221 (56.7%)	14 (17.5%)	19 (20.2%)	188 (87.0%)
I am confident that public authorities decide in the best interest of the community:				
Neutral	92 (23.2%)	38 (46.3%)	29 (29.9%)	25 (11.5%)
No	73 (18.4%)	17 (20.7%)	42 (43.3%)	14 (6.45%)
Yes	231 (58.3%)	27 (32.9%)	26 (26.8%)	178 (82.0%)
Influenza vaccination is unnecessary because flu is not common anymore:				
Neutral	97 (24.9%)	38 (47.5%)	28 (29.5%)	31 (14.5%)
No	252 (64.8%)	40 (50.0%)	50 (52.6%)	162 (75.7%)
Yes	40 (10.3%)	2 (2.50%)	17 (17.9%)	21 (9.81%)
My immune system is so strong, it also protects me against flu:				
Neutral	91 (23.4%)	37 (46.2%)	20 (21.1%)	34 (15.9%)
No	210 (54.0%)	34 (42.5%)	42 (44.2%)	134 (62.6%)
Yes	88 (22.6%)	9 (11.2%)	33 (34.7%)	46 (21.5%)
Flu infection is not so severe that I should be vaccinated:				
Neutral	88 (22.4%)	21 (26.2%)	31 (32.3%)	36 (16.7%)
No	230 (58.7%)	49 (61.3%)	34 (35.4%)	147 (68.1%)
Yes	74 (18.9%)	10 (12.5%)	31 (32.3%)	33 (15.3%)
Everyday stress will prevent me from getting vaccinated against influenza:				
Neutral	85 (21.6%)	28 (34.1%)	30 (31.9%)	27 (12.4%)
No	264 (67.0%)	46 (56.1%)	52 (55.3%)	166 (76.1%)
Yes	45 (11.4%)	8 (9.76%)	12 (12.8%)	25 (11.5%)
It is inconveniencing for me to receive vaccinations against influenza:				
Neutral	85 (21.5%)	32 (39.5%)	27 (27.8%)	26 (11.9%)
No	235 (59.3%)	44 (54.3%)	42 (43.3%)	149 (68.3%)
Yes	76 (19.2%)	5 (6.17%)	28 (28.9%)	43 (19.7%)
Visiting the vaccination clinic will make me feel uncomfortable, and this will keep me from getting vaccinated against influenza:				
Neutral	66 (16.8%)	25 (30.5%)	21 (21.9%)	20 (9.35%)
No	237 (60.5%)	34 (41.5%)	47 (49.0%)	156 (72.9%)
Yes	89 (22.7%)	23 (28.0%)	28 (29.2%)	38 (17.8%)
When I think about getting vaccinated against influenza, I weigh benefits and risks to make the best decision possible:				
Neutral	64 (16.1%)	22 (26.8%)	18 (18.6%)	24 (11.0%)
No	77 (19.3%)	11 (13.4%)	20 (20.6%)	46 (21.0%)
Yes	257 (64.6%)	49 (59.8%)	59 (60.8%)	149 (68.0%)
For every influenza vaccine dose, I will closely consider whether it is useful for me:				
Neutral	81 (20.6%)	29 (35.4%)	27 (28.1%)	25 (11.6%)
No	74 (18.8%)	11 (13.4%)	17 (17.7%)	46 (21.4%)
Yes	238 (60.6%)	42 (51.2%)	52 (54.2%)	144 (67.0%)
It is important for me to fully understand the topic of vaccination before I get vaccinated against influenza:				
Neutral	60 (15.2%)	24 (29.6%)	19 (19.6%)	17 (7.87%)
No	55 (14.0%)	8 (9.88%)	12 (12.4%)	35 (16.2%)
Yes	279 (70.8%)	49 (60.5%)	66 (68.0%)	164 (75.9%)
When everyone is vaccinated against influenza, I don’t have to get vaccinated, too:				
Neutral	81 (21.2%)	28 (35.4%)	37 (41.1%)	16 (7.51%)
No	183 (47.9%)	16 (20.3%)	23 (25.6%)	144 (67.6%)
Yes	118 (30.9%)	35 (44.3%)	30 (33.3%)	53 (24.9%)
I will get vaccinated against influenza because I will be protecting people with a weaker immune system:				
Neutral	75 (19.2%)	27 (33.3%)	33 (34.7%)	15 (6.98%)
No	60 (15.3%)	12 (14.8%)	22 (23.2%)	26 (12.1%)
Yes	256 (65.5%)	42 (51.9%)	40 (42.1%)	174 (80.9%)
Vaccination is a collective responsibility to prevent the spread of diseases like influenza:				
Neutral	70 (17.8%)	29 (35.4%)	28 (28.6%)	13 (6.10%)
No	33 (8.40%)	3 (3.66%)	14 (14.3%)	16 (7.51%)
Yes	290 (73.8%)	50 (61.0%)	56 (57.1%)	184 (86.4%)

Values shown are absolute counts (percentages).

**Table 3 vaccines-10-01176-t003:** Univariate and multivariable logistic regression analysis of factors associated with influenza vaccine acceptance (vs. hesitancy) among healthcare professionals at the start of COVID-19 vaccine rollout in Cape Town, South Africa. The table contains six different multivariable models (six subheadings), the first of which assesses sociodemographic factors (including religion), while the second through sixth models assess the effect of variables related to specific items pertaining to the 5C questionnaire components (i.e., confidence, complacency, constraint, calculation, and collective responsibility) that are independently associated with influenza vaccination acceptance.

Variables		Influenza Vaccine Intention		Acceptance versus Hesitancy	
		Hesitancy	Acceptance	cOR (95% CI, *p*-Value)	aOR (95% CI, *p*-Value)
Sociodemographic characteristics					
Age groups	18–24 years	31 (77.5)	9 (22.5)	-	-
	25–34 years	48 (46.2)	56 (53.8)	4.02 (1.80–9.74, *p* = 0.001)	3.31 (1.25–9.97, *p* = 0.02)
	35–44 years	41 (42.7)	55 (57.3)	4.62 (2.05–11.29, *p* < 0.001)	4.27 (1.60–12.93, *p* = 0.01)
	45–54 years	26 (38.8)	41 (61.2)	5.43 (2.30–13.84, *p* < 0.001)	6.54 (2.26–21.27, *p* = 0.001)
	55–78 years	9 (21.4)	33 (78.6)	12.63 (4.64–38.09, *p* < 0.001)	9.53 (2.55–40.32, *p* = 0.001)
Gender	Male	61 (44.9)	75 (55.1)	-	-
	Female	112 (45.3)	135 (54.7)	0.98 (0.64–1.49, *p* = 0.93)	1.36 (0.77–2.43, *p* = 0.29)
Health worker role	Admin support	19 (37.3)	32 (62.7)	-	-
	Nurses	107 (55.4)	86 (44.6)	0.48 (0.25–0.89, *p* = 0.022)	0.48 (0.21–1.09, *p* = 0.08)
	Other health workers	49 (49.5)	50 (50.5)	0.61 (0.30–1.20, *p* = 0.155)	0.64 (0.26–1.56, *p* = 0.32)
	Physicians	3 (6.1)	46 (93.9)	9.10 (2.81–41.08, *p* = 0.001)	24.43 (3.97–480.39, *p* = 0.004)
Highest educational level	High School Graduate	102 (54.3)	86 (45.7)	-	-
	Below High School	3 (27.3)	8 (72.7)	3.16 (0.88–14.78, *p* = 0.10)	2.81 (0.65–15.15, *p* = 0.19)
	University	64 (35.2)	118 (64.8)	2.19 (1.44–3.33, *p* < 0.001)	1.83 (1.01–3.37, *p* = 0.048)
Personal income	Less than R10,000 per month	40 (51.3)	38 (48.7)	-	-
	More than R50,000 per month	10 (18.9)	43 (81.1)	4.53 (2.06–10.70, *p* < 0.001)	0.53 (0.15–1.90, *p* = 0.32)
	R10,000–R50,000 per month	118 (50.2)	117 (49.8)	1.04 (0.62–1.75, *p* = 0.87)	1.01 (0.48–2.10, *p* = 0.98)
Religion	African or Hindu or Jewish	11 (47.8)	12 (52.2)	-	-
	Christian	119 (45.4)	143 (54.6)	1.10 (0.46–2.60, *p* = 0.82)	1.69 (0.57–5.18, *p* = 0.34)
	Muslim	33 (61.1)	21 (38.9)	0.58 (0.21–1.56, *p* = 0.28)	0.79 (0.21–2.92, *p* = 0.72)
	None	8 (29.6)	19 (70.4)	2.18 (0.69–7.18, *p* = 0.19)	1.27 (0.32–5.24, *p* = 0.74)
Variables related to confidence in influenza vaccines					
Received influenza vaccine in the past	No	134 (69.8)	58 (30.2)	-	-
	Yes	45 (22.5)	155 (77.5)	7.96 (5.10–12.63, *p* < 0.001)	6.41 (3.15–13.53, *p* < 0.001)
Compatible with religion	Neutral	62 (72.1)	24 (27.9)	-	-
	No	73 (73.0)	27 (27.0)	0.96 (0.50–1.83, *p* = 0.89)	0.92 (0.28–2.99, *p* = 0.89)
	Yes	45 (21.2)	167 (78.8)	9.59 (5.47–17.31, *p* < 0.001)	1.06 (0.35–3.03, *p* = 0.91)
Vaccines are safe	Neutral	73 (79.3)	19 (20.7)	-	-
	No	72 (86.7)	11 (13.3)	0.59 (0.25–1.30, *p* = 0.20)	0.68 (0.15–2.81, *p* = 0.61)
	Yes	34 (15.3)	188 (84.7)	21.24 (11.62–40.56, *p* < 0.001)	2.31 (0.65–7.97, *p* = 0.19)
Vaccines are effective	Neutral	78 (83.0)	16 (17.0)	-	-
	No	63 (84.0)	12 (16.0)	0.93 (0.40–2.10, *p* = 0.86)	0.91 (0.23–3.77, *p* = 0.90)
	Yes	33 (14.9)	188 (85.1)	27.77 (14.82–54.94, *p* < 0.001)	5.82 (1.85–18.86, *p* = 0.003)
Authorities have best interest	Neutral	67 (72.8)	25 (27.2)	-	-
	No	59 (80.8)	14 (19.2)	0.64 (0.30–1.32, *p* = 0.232)	0.52 (0.16–1.65, *p* = 0.27)
	Yes	53 (22.9)	178 (77.1)	9.00 (5.25–15.88, *p* < 0.001)	1.72 (0.63–4.46, *p* = 0.28)
Variables related to complacency towards influenza vaccines					
Vaccination is unnecessary	Neutral	66 (68.0)	31 (32.0)	-	-
	No	90 (35.7)	162 (64.3)	3.83 (2.35–6.37, *p* < 0.001)	3.58 (1.83–7.22, *p* < 0.001)
	Yes	19 (47.5)	21 (52.5)	2.35 (1.11–5.04, *p* = 0.03)	4.44 (1.65–12.41, *p* = 0.004)
Have strong immune system	Neutral	57 (62.6)	34 (37.4)	-	-
	No	76 (36.2)	134 (63.8)	2.96 (1.79–4.96, *p* < 0.001)	2.05 (1.06–3.99, *p* = 0.03)
	Yes	42 (47.7)	46 (52.3)	1.84 (1.02–3.35, *p* = 0.046)	1.71 (0.78–3.81, *p* = 0.18)
Flu is not severe enough	Neutral	52 (59.1)	36 (40.9)	-	-
	No	83 (36.1)	147 (63.9)	2.56 (1.55–4.26, *p* < 0.001)	1.23 (0.62–2.42, *p* = 0.54)
	Yes	41 (55.4)	33 (44.6)	1.16 (0.62–2.18, *p* = 0.64)	0.55 (0.23–1.29, *p* = 0.17)
Variables related to constraints affecting uptake of influenza vaccines					
Everyday stress	Neutral	58 (68.2)	27 (31.8)	-	-
	No	98 (37.1)	166 (62.9)	3.64 (2.18–6.20, *p* < 0.001)	2.05 (1.02–4.18, *p* = 0.046)
	Yes	20 (44.4)	25 (55.6)	2.69 (1.28–5.72, *p* = 0.009)	3.45 (1.29–9.67, *p* = 0.016)
Inconvenient to be vaccinated	Neutral	59 (69.4)	26 (30.6)	-	-
	No	86 (36.6)	149 (63.4)	3.93 (2.33–6.78, *p* < 0.001)	1.66 (0.81–3.42, *p* = 0.17)
	Yes	33 (43.4)	43 (56.6)	2.96 (1.56–5.71, *p* = 0.001)	1.85 (0.79–4.37, *p* = 0.16)
Clinic visits uncomfortable	Neutral	46 (69.7)	20 (30.3)	-	-
	No	81 (34.2)	156 (65.8)	4.43 (2.49–8.13, *p* < 0.001)	3.55 (1.78–7.27, *p* < 0.001)
	Yes	51 (57.3)	38 (42.7)	1.71 (0.88–3.40, *p* = 0.116)	1.45 (0.64–3.36, *p* = 0.38)
Variables related to risk calculation with respect to influenza vaccines					
Weigh benefits and risks	Neutral	40 (62.5)	24 (37.5)	-	-
	No	31 (40.3)	46 (59.7)	2.47 (1.26–4.94, *p* = 0.01)	1.19 (0.51–2.77, *p* = 0.69)
	Yes	108 (42.0)	149 (58.0)	2.30 (1.32–4.08, *p* = 0.004)	1.20 (0.59–2.45, *p* = 0.61)
Consider usefulness of each dose	Neutral	56 (69.1)	25 (30.9)	-	-
	No	28 (37.8)	46 (62.2)	3.68 (1.91–7.26, *p* < 0.001)	2.62 (1.17–5.98, *p* = 0.02)
	Yes	94 (39.5)	144 (60.5)	3.43 (2.02–5.96, *p* < 0.001)	2.80 (1.42–5.65, *p* = 0.003)
Need to fully understand	Neutral	43 (71.7)	17 (28.3)	-	-
	No	20 (36.4)	35 (63.6)	4.43 (2.05–9.92, *p* < 0.001)	2.97 (1.20–7.58, *p* = 0.02)
	Yes	115 (41.2)	164 (58.8)	3.61 (1.99–6.79, *p* < 0.001)	2.09 (1.00–4.50, *p* = 0.05)
Variables associated with collective responsibility for influenza vaccines					
Everyone vaccinated not me	Neutral	65 (80.2)	16 (19.8)	-	-
	No	39 (21.3)	144 (78.7)	15.00 (8.00–29.58, *p* < 0.001)	6.96 (3.24–15.50, *p* < 0.001)
	Yes	65 (55.1)	53 (44.9)	3.31 (1.75–6.54, *p* < 0.001)	1.56 (0.72–3.51, *p* = 0.27)
Vaccinated to protect weaker immunity	Neutral	60 (80.0)	15 (20.0)	-	-
	No	34 (56.7)	26 (43.3)	3.06 (1.44–6.68, *p* = 0.004)	1.55 (0.57–4.33, *p* = 0.40)
	Yes	82 (32.0)	174 (68.0)	8.49 (4.66–16.33, *p* < 0.001)	3.44 (1.56–7.95, *p* = 0.003)
Collective action	Neutral	57 (81.4)	13 (18.6)	-	-
	No	17 (51.5)	16 (48.5)	4.13 (1.68–10.47, *p* = 0.002)	2.98 (0.86–10.70, *p* = 0.09)
	Yes	106 (36.6)	184 (63.4)	7.61 (4.10–15.12, *p* < 0.001)	2.87 (1.23–7.07, *p* = 0.017)

Values shown are absolute counts (percentages) and odds ratios (95% confidence intervals and *p* values). The category “hesitancy” includes participants who were unsure whether they would take the influenza vaccine during the flu season and those who planned to refuse the influenza vaccine.

## Data Availability

The datasets used and/or analyzed during the current study are available from the corresponding author on reasonable request.

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
