# Peer review of "Influenza Vaccination Uptake and Hesitancy among Healthcare Workers in Early 2021 at the Start of the COVID-19 Vaccine Rollout in Cape Town, South Africa"

_vaccines, 2022, doi:10.3390/vaccines10081176_

Round 1
Reviewer 1 Report
This study by Alobwede and colleagues is important as vaccination is the most effective preventive measure against influenza, and being healthcare workers in regular contact with ill patients, they represent a priority target group for receiving the vaccination.
This study carries important findings, even if several studies have already published data from different countries where they have documented the lack of influenza vaccination in healthcare workers. This paper is not novel in that sense. However, this information is important also to understand that psychological, behavioral, and contextual factors play a role in deciding to get vaccinated or not.
There are limitations that should be considered.
- Some sentences in the introduction are not clear, they need to be rephrased (e.g. "Despite the severity of influenza, there are safe vaccines available albeit a low uptake (hesitancy) among specific risk groups resulting in the high burden of flu pandemic" or "Despite these recommendations and reported efficacy of HCWs influenza vaccination in reducing sickness, absenteeism rates mostly in the high-income parts of the world, with annual vaccination rates varying between 44% and 54% in the UK".
- In the introduction, you mention the fact that reports show that influenza vaccination does not only protect against influenza but also reduces the risk of COVID-19 infection...[...]. This is currently debated as the plausibility of this assumption has not been clearly demonstrated yet (with immunological studies); moreover results are not consistent across different studies, and studies present many biases (below are some references).
- https://www.mdpi.com/2076-393X/10/4/606
- https://www.mdpi.com/1660-4601/17/21/7870
- https://pubmed.ncbi.nlm.nih.gov/34989811/
The only thing we can assume for "sure" is that influenza vaccination does not increase the risk of SARS-CoV-2 infection (or worse COVID-19 outcomes).
- https://www.ncbi.nlm.nih.gov/pmc/articles/PMC7437565/
- The methodology section is well described and written. I suggest specifically explaining how the participants were recruited (were they coming from the same hospital? Have you planned to recruit any HCW or did you want a certain proportion of doctors, nurses, etc? Why?)
- Was the questionnaire validated?
- I liked the way you dichotomized the outcome variable (1-4 vs 5-7) and I find it appropriate, also considering other studies that were published and included the "neutral" subjects in the hesitant group.
- I suggest extending and better explaining the table descriptions. Table 3 is a bit complicated and it would be handy for a reader to have a better explanation on top of the table, with rows and columns explained.
- In the results, you mention that "In addition, HCWs who believe that influenza vaccines are effective were more likely to accept the vaccines with an acceptance frequency of 85.1% versus 14.9% hesitancy (OR 5.82, 95% CI 1.85-18.86, p = 0.003) as compared to those with neutral position." This is a quite important result. It is often reported by many studies, but poorly discussed. I suggest better discussing this finding in the discussion, also mentioning the importance of taking action to better study IV effectiveness on one hand and explaining it to the HCWs on the other.
https://pubmed.ncbi.nlm.nih.gov/33207626/
- In my opinion, the discussion section should have a paragraph (just one, with all information in it) with a specific focus on the impact of the pandemic on influenza vaccination coverage rates in HCWs. I suggest collating all the evidence, explaining what happened in your sample and comparing it to different studies you found in the literature. Also, I think you should discuss it in light of what happened with different age groups (e.g. the elderly, as the pandemic seemed to have been followed by an increase in VCRs in this group).
https://pubmed.ncbi.nlm.nih.gov/34915972/
https://www.ncbi.nlm.nih.gov/pmc/articles/PMC9026798/
- I suggest reorganising the discussion section and summarising it a tad, as it is long and sometimes difficult to follow.
Author Response
Response to Reviewer 1 Comments
This study by Alobwede and colleagues is important as vaccination is the most effective preventive measure against influenza, and being healthcare workers in regular contact with ill patients, they represent a priority target group for receiving the vaccination.
This study carries important findings, even if several studies have already published data from different countries where they have documented the lack of influenza vaccination in healthcare workers. This paper is not novel in that sense. However, this information is important also to understand that psychological, behavioral, and contextual factors play a role in deciding to get vaccinated or not.
There are limitations that should be considered.
Introduction
Point 1: Some sentences in the introduction are not clear, they need to be rephrased (e.g. “Despite the severity of influenza, there are safe vaccines available albeit a low uptake (hesitancy) among specific risk groups resulting in the high burden of flu pandemic” or “Despite these recommendations and reported efficacy of HCWs influenza vaccination in reducing sickness, absenteeism rates mostly in the high-income parts of the world, with annual vaccination rates varying between 44% and 54% in the UK”.
Response 1: This sentence has been clarified in the manuscript
Despite the severity of influenza, there are safe vaccines available albeit a low uptake (i.e., vaccine hesitancy) among specific risk groups which results in high burden of influenza infections.
This sentence has been clarified in the manuscript
Despite these recommendations and known efficacy of influenza vaccines in reducing infection and severity of sickness, HCW absenteeism from work due to flu remains high even in the high-income parts of the world [11,12]. This is associated with low annual vaccination rate, for example in the UK, influenza vaccination coverage varies between 44% and 54% [11].
Point 2: In the introduction, you mention the fact that reports show that influenza vaccination does not only protect against influenza but also reduces the risk of COVID-19 infection...[...]. Kong, Gwyneth, et al Del Riccio M,et .al This is currently debated as the plausibility of this assumption has not been clearly demonstrated yet (with immunological studies); moreover results are not consistent across different studies, and studies present many biases (below are some references).
- https://www.mdpi.com/2076-393X/10/4/606: Effect of COVID-19 Pandemic on Influenza Vaccination Intention: A Meta-Analysis and Systematic Review_vaccines-10-00606-v2
- https://www.mdpi.com/1660-4601/17/21/7870: The Association between Influenza Vaccination and the Risk of SARS-CoV-2 Infection, Severe Illness, and Death: A Systematic Review of the Literature_ijerph-17-07870
- https://pubmed.ncbi.nlm.nih.gov/34989811/: Effect of influenza vaccination on risk of COVID-19: A prospective cohort study of 46,000 health care workers_jiac001
The only thing we can assume for “sure” is that influenza vaccination does not increase the risk of SARS-CoV-2 infection (or worse COVID-19 outcomes).
- https://www.ncbi.nlm.nih.gov/pmc/articles/PMC7437565/: The impact of influenza vaccination on the COVID-19 pandemic? Evidence and lessons for public health policies
https://pubmed.ncbi.nlm.nih.gov/34915972/: Letter to the editor: Increase of influenza vaccination coverage rates during the COVID-19 pandemic and implications for the upcoming influenza season in northern hemisphere countries and Australia_eurosurv-26-50-9
Response 2: This has been clarified in the manuscript
Existing literature suggest that influenza vaccination influences COVID-19 infection, however, with inconsistent results across different studies.
Moreover, Moreover, there is no clarity on the association of influenza vaccination on COVID-19 infection. Kong and colleagues conducted a systematic review of 27 studies from 1 January 2019 to 31 December 2021 with the aim to evaluate the “Effect of COVID-19 Pandemic on Influenza Vaccination Intention” [20]. Finding from a systematic synthesis of 27 studies on the effects of COVID-19 pandemic on influenza showed an increase in intention to vaccinate in 2020/2021 and this was regardless of the participants demographics [20]. Thus, despite inconsistent results, this suggested that COVID-19 pandemic increased the intention to vaccinate against influenza in other parts of the world.
We would wonder if the COVID-19 pandemic driven increase in influenza vaccination would influence SARS-CoV-2 infection? Another systematic review in 2020 by Del Riccio M and colleagues assessed the association between influenza vaccination and the risk of SARS-CoV-2 infection, severe illness, and death [21]. The result of another systematic synthesis found no evidence of a significant increase in the risk of infection or in the se-verity illness or lethality, with some rather reporting significantly inverse associations [21]. Hence, the findings are supporting measures that are aimed at raising HCWs influenza vaccination coverage in the COVID-19 era. Also, in a most recent study in 2022 by Kristensen and colleagues, they assessed in a prospective cohort study of 46,000 HCWs if their intention towards influenza vaccination had an impact on the risk of COVID-19 [22].
Further studies by Kristensen and colleagues also found that influenza vaccination did not affect the risk of SARS-CoV-2 infection or COVID-19. In summary, evidence so far suggests that influenza vaccination does not increase the risk of SARS-CoV-2 infection (or worse COVID-19 outcomes). Thus, understanding the impact of influenza vaccination on the risk of SARS-CoV-2 infection or COVID-19 on HCWs is critical and would be of great importance to add to the evidence pool to guide public health policies for the future [23,24].
Methods
Point 3: The methodology section is well described and written. I suggest specifically explaining how the participants were recruited (were they coming from the same hospital? Have you planned to recruit any HCW or did you want a certain proportion of doctors, nurses, etc? Why?)
Response 3: This has been responded to in the Editors comments as well (see above)
Thank you for the compliment.
The study was a cross sectional survey of all HCWs 18 years and older of all races and gender living and working in hospitals and other healthcare settings in Cape Town, South Africa. We employed a convenience sampling and participants were HCWs from both private and government facilities who participated on voluntary basis. These included nurses, physicians, pharmacists, hospital administrative personals, health researchers, and radiologist.
There was no specific intention to recruit HCWs of a particular role.
Point 4: Was the questionnaire validated?
Response 4: This has been responded to in the Editors comments as well (see above)
Yes, …this study adapted standardized 15 questions published by Betsch and colleagues in 2018 [49] on five psychological antecedents of vaccinations namely constraints, confidence, calculation of risk, complacency, and collective responsibility (also known as the 5C tool) [49], and further contextualized in our setting. In addition, questions were asked regarding HCWs religion being compatible with influenza vaccination. The validation of the 5C tool and questionnaire for use in our setting is a subject of another manuscript currently in preparation.
Point 5: I liked the way you dichotomized the outcome variable (1-4 vs 5-7) and I find it appropriate, also considering other studies that were published and included the “neutral” subjects in the hesitant group.
Response 5: Thank you for the compliment
Point 6: I suggest extending and better explaining the table descriptions. Table 3 is a bit complicated and it would be handy for a reader to have a better explanation on top of the table, with rows and columns explained.
Response 6: The table description has been explained better as recommended by the reviewer.
Table 1. Sociodemographic features of 401 healthcare professionals in Cape Town, South Africa, stratified by their intention to accept, decline, or remain neutral once the influenza vaccination is available to them during Covid-19 pandemic. Survey period coincides with the start of COVID-19 vaccine rollout.
Table 2. Attitudes toward influenza vaccination of 401 healthcare professionals in Cape Town, South Africa, using the 5C questionnaire items (with the inclusion of religious influence), stratified by their intention to accept, decline, or remain neutral once influenza vaccine is available to them during Covid-19 pandemic. Survey period coincides with the start of COVID-19 vaccine rollout.
Table 3. Univariate and multivariable logistic regression analysis of factors associated with influenza vaccine acceptance (vs hesitancy) among healthcare professionals at the start of COVID-19 vaccine rollout in Cape Town, South Africa: The table contains six different multivariable models (six subheadings), the first of which assesses sociodemographic factors (including religion) while the second through sixth models assess the effect of variables related to specific items pertaining to the 5C questionnaire components (i.e. confidence, complacency, constraint, calculation and collective responsibility) that are independently associated with influenza vaccination acceptance.
Results
Point 7: In the results, you mention that “In addition, HCWs who believe that influenza vaccines are effective were more likely to accept the vaccines with an acceptance frequency of 85.1% versus 14.9% hesitancy (OR 5.82, 95% CI 1.85-18.86, p = 0.003) as compared to those with neutral position.” This is a quite important result. It is often reported by many studies, but poorly discussed. I suggest better discussing this finding in the discussion, also mentioning the importance of taking action to better study IV effectiveness on one hand and explaining it to the HCWs on the other.
https://pubmed.ncbi.nlm.nih.gov/33207626/: Attitudes and Perception of Healthcare Workers Concerning Influenza Vaccination during the 20192020 Season: A Survey of Sicilian University Hospitals_vaccines-08-00686
Costantino C, Ledda C, Squeri R, Restivo V, Casuccio A, Rapisarda V, Graziano G, Alba D, Cimino L, Conforto A, Costa GB, D’Amato S, Mazzitelli F, Vitale F, Genovese C. Attitudes and Perception of Healthcare Workers Concerning Influenza Vaccination during the 2019/2020 Season: A Survey of Sicilian University Hospitals. Vaccines (Basel). 2020 Nov 16;8(4):686. Doi: 10.3390/vaccines8040686. PMID: 33207626; PMCID: PMC7711679.
Response 7: This had been discussed in the manuscript and a further statement has been added.
This finding from our study could also be explained by findings in the study by Costantino and colleagues performed in three Sicilian University Hospitals assessing influenza vaccination adherence or refusal, and to understand the attitudes and perceptions of vaccinated HCWs during the 2019/2020 influenza season [61]. They reported that in the total of 2356 vaccinated HCWs that answered the questionnaire, the main reason for influenza vaccination adherence was to protect patients. Also, higher self-perceived risk of contracting influenza and a positive attitude to recommending vaccination to patients were significantly associated with influenza vaccination adherence during the last five seasons as reported after multivariable analysis. However, they found that fear of an adverse reaction was the main reason for influenza vaccine refusal. One key difference between both studies was the larger sample size in the study in Italy compared to 414 in our study. In addition, several studies conducted in Canada [31], Ghana [36], UK [33], Israel [34], and Lebanon [37] have reported that HCWs confidence to accept influenza vaccine led to them recommending the vaccine to their co-workers, patients, or patients’ families than those who did not intend to be vaccinated.
Discussion
Point 8: In my opinion, the discussion section should have a paragraph (just one, with all information in it) with a specific focus on the impact of the pandemic on influenza vaccination coverage rates in HCWs. I suggest collating all the evidence, explaining what happened in your sample and comparing it to different studies you found in the literature. Also, I think you should discuss it in light of what happened with different age groups (e.g. the elderly, as the pandemic seemed to have been followed by an increase in VCRs in this group).
https://www.ncbi.nlm.nih.gov/pmc/articles/PMC9026798/: Effect of COVID-19 Pandemic on Influenza Vaccination Intention_A Meta-Analysis and Systematic Review_vaccines-10-00606
https://pubmed.ncbi.nlm.nih.gov/34915972/: Letter to the editor: Increase of influenza vaccination coverage rates during the COVID-19 pandemic and implications for the upcoming influenza season in northern hemisphere countries and Australia_eurosurv-26-50-9
Response 8: The discussion does have four paragraphs that discuss the impact of the COVID-19 pandemic on influenza vaccine coverage in HCWs as it is such an important aspect of the study. We will include some highlighted studies by the reviewer.
Kong et al. conducted a meta-analysis and systematic review of 27 studies from 1 January 2019 to 31 December 2021 with the aim to evaluate the “Effect of COVID-19 Pandemic on Influenza Vaccination Intention” [20]. The finding showed that studies reported increased intention to vaccinate in 2020/21 and this was regardless of the demographics. In accordance with our study HCWs who believed that influenza vaccines are effective were more likely to accept the vaccines as compared to those with neutral position and there is evidence of increase influenza vaccination coverage rates during the COVID-19 pandemic [24]. In addition, older HCWs were most likely to receive influenza vaccines compared to younger colleagues. Overall, this could be supported by the fact that HCWs confidence to accept influenza vaccine would lead to them recommending the vaccine to their co-workers, patients, or patients’ families than those who did not intend to be vaccinated as reported in several studies carried out in Canada [31], UK [33], Israel [34].
Point 9: I suggest reorganising the discussion section and summarising it a tad, as it is long and sometimes difficult to follow.
Response 9: The discussion has been updated.
Reviewer 2 Report
Authors, this is a very through study on parameters that impact influenza virus vaccine hesitancy in health care workers in South Africa. The main limitation of this study is a very select group of participants in a geographically limited area of South Africa and only a limited timepoint. This is an interesting and important study as vaccine acceptance among health care workers can have a significant effect on acceptance of vaccines by the public. Thus, determining what influences vaccine hesitancy among health care workers can hopefully provide for better educational approaches to increase the acceptance of influenza vaccines by the health care industry.
The methods used in this study are appropriate, there was a sufficient number of participants to allow for statistical analysis and the conclusions reached by the authors are supported by the data.
On page 12, the next to the last paragraph, you found overall low influenza vaccine rates similar to a previous report during the COVID-19 pandemic. Is it possible that the lower influenza rate during the pandemic may have been due to 1) rates of influenza were lower during the pandemic, thus not considered a priority; 2) fear of visiting medical clinics as this might increase one's chances for contacting COVID19 or perhaps it was just not a priority in fact of a new respiratory virus infection?
In your conclusion, perhaps one approach to mitigating vaccine hesitancy among health care workers is to include greater emphasis on vaccines and viral diseases in the medical curriculum. Often the time given to infectious diseases in some medical training is very brief. Perhaps more detailed teaching and information on vaccines, their efficacy, immune responses, etc could be beneficial in reducing this hesitancy?
Author Response
Response to Reviewer 2 Comments
Authors, this is a very through study on parameters that impact influenza virus vaccine hesitancy in health care workers in South Africa. The main limitation of this study is a very select group of participants in a geographically limited area of South Africa and only a limited timepoint. This is an interesting and important study as vaccine acceptance among health care workers can have a significant effect on acceptance of vaccines by the public. Thus, determining what influences vaccine hesitancy among health care workers can hopefully provide for better educational approaches to increase the acceptance of influenza vaccines by the health care industry.
Point 1: The methods used in this study are appropriate, there was a sufficient number of participants to allow for statistical analysis and the conclusions reached by the authors are supported by the data.
Response 1: Thank you very much for the compliments. We have also acknowledged the caution for generalisability under the limitation section.
Point 2: On page 12, the next to the last paragraph, you found overall low influenza vaccine rates similar to a previous report during the COVID-19 pandemic. Is it possible that the lower influenza rate during the pandemic may have been due to 1) rates of influenza were lower during the pandemic, thus not considered a priority; 2) fear of visiting medical clinics as this might increase one's chances for contacting COVID19 or perhaps it was just not a priority in fact of a new respiratory virus infection?
Response 2: That is a good suggestion by the reviewer. It is true that influenza vaccine coverage was suboptimum and was low. More measures should be put in place to encourage HCWs influenza vaccination. We will acknowledge these under study the limitations and suggest way forward.
Limitation
It is possible that the lower influenza rate during the pandemic was due to 1) the fact that influenza rates were lower during the pandemic and were therefore not considered a priority; 2) the fear of visiting medical clinics due to the possibility of contracting COVID-19; or 3) the fact that a new respiratory virus infection was instead becoming a priority. Future research may explore correlating the number of monthly influenza cases with vaccination hesitancy, as well as analysing vaccine hesitancy in a prospective cohort accounting for COVID-19 high vs low community’ transmission period.
Point 3: In your conclusion, perhaps one approach to mitigating vaccine hesitancy among health care workers is to include greater emphasis on vaccines and viral diseases in the medical curriculum. Often the time given to infectious diseases in some medical training is very brief. Perhaps more detailed teaching and information on vaccines, their efficacy, immune responses, etc could be beneficial in reducing this hesitancy?
Response 3:
Conclusion
For future flu seasons the importance of tailored education programs targeting younger HCWs, continues heath education and campaigns on vaccines uptake and more information about the content of flu vaccines would be vital to improve vaccine uptake. In addition, emphasis on inclusion of education on vaccines and viral diseases in the medical curriculum as one approach to mitigating vaccine hesitancy among HCWs should be encouraged . This would help alleviate fears and provide avenues to engage communities, assist and address the pertinent issues and build influenza vaccine confidence among HCWs and reassure them about the effectiveness and safety of the seasonal flu vaccines. Finally, we believe that this paper provides the scientific evidence on influenza vaccine acceptance among HCWs in South Africa and broadly the sub-Saharan African region. It further provides better understanding of influenza vaccine hesitancy among HCWs in the time of the COVID-19 pandemic.
Round 2
Reviewer 1 Report
In the revised version, the manuscript has improved in readability and overall quality. The authors made significant efforts to address all the comments.
Particularly, the purpose has been made more clear, the literature review updated to be current, and the way to pursue the research aim has been clearly explained.